# Development of Instrumented Running Prosthetic Feet for the Collection of Track Loads on Elite Athletes [note 1]

**DOI:** 10.3390/s20205758

**Published:** 2020-10-10

**Authors:** Nicola Petrone, Gianfabio Costa, Gianmario Foscan, Antonio Gri, Leonardo Mazzanti, Gianluca Migliore, Andrea Giovanni Cutti

**Affiliations:** 1Department of Industrial Engineering, University of Padova, Via Venezia 1, 35131 Padova, Italy; gianfabio.costa@gmail.com (G.C.); gianmario.foscan@gmail.com (G.F.); antonio.gri@studenti.unipd.it (A.G.); leomaz800@gmail.com (L.M.); 2INAIL, Via Rabuina 14, 40054 Vigorso di Budrio, Italy; gi.migliore@inail.it (G.M.); ag.cutti@inail.it (A.G.C.)

**Keywords:** running specific prostheses (RSP), running prosthetic feet (RPF), strain gauges, calibration, validation, structural loads, inertial sensors, sprint, long jump

## Abstract

Knowledge of loads acting on running specific prostheses (RSP), and in particular on running prosthetic feet (RPF), is crucial for evaluating athletes’ technique, designing safe feet, and biomechanical modelling. The aim of this work was to develop a J-shaped and a C-shaped wearable instrumented running prosthetic foot (iRPF) starting from commercial RPF, suitable for load data collection on the track. The sensing elements are strain gauge bridges mounted on the foot in a configuration that allows decoupling loads parallel and normal to the socket-foot clamp during the stance phase. The system records data on lightweight athlete-worn loggers and transmits them via Wi-Fi to a base station for real-time monitoring. iRPF calibration procedure and static and dynamic validation of predicted ground-reaction forces against those measured by a force platform embedded in the track are reported. The potential application of this wearable system in estimating determinants of sprint performance is presented.

## 1. Introduction

Running has become one of the most spectacular competitions at the Paralympic games: sprint running and long jumping on prosthetic carbon blades have reached such high performance level to raise discussion about fairness of athletes with prostheses against abled-bodied [1]. On the other hand, attention and enthusiasm for such fascinating disciplines have increased the awareness of persons with amputation and improved inclusiveness by attracting many to the practice of sports.

Since the introduction of prosthetic feet for running, researchers focused not only on the properties of running prosthetic feet (RPF) but also on the nature of loads that the ground applies through the foot to the leg and eventually to athlete’s body [2,3,4,5]. A vast literature exists on sprint running and long jumping, mainly for able-bodied than Paralympic athletes. In several studies that focused on sprint performance, researchers measured the ground reaction forces (GRF) through a set of force platforms embedded in the track [6,7]. Most of these works were carried out indoor, on elite athletes performing their sprint or jumps takeoff on force platforms. In particular, three approaches are worth noting.

In the first approach, the GRF were captured together with athlete’s kinematics on a single or double step over 3–7 platforms [2,7].

The study of Nagahara et al. [6] is representative of the second approach, in which a much larger number of platforms is used. Specifically, Nagahara adopted over 50 force platforms and 60 motion capture cameras installed in the runway of an indoor track while measuring the sprint kinematics and kinetics of abled-bodied sprinters.

The third approach to be considered is the use of an instrumented treadmill equipped with force platforms [8]. All approaches have pros and cons: high reliability of loads and easy integration with motion capture are advantages of the first approach [2,7], despite limitations consisting in collecting only one step, running in an indoor environment and facing the risk of collecting "bad steps" as sprinters may aim to hit the platform. In the second approach [6], despite the advantage collecting multiple steps from a natural sprint, a very extensive and expensive installation is needed and this can rarely be afforded. Finally, the third approach [8] combines the ability of measuring running GRFs of both sides with high reliability usually integrated with motion capture on multiple stationary steps: on the other hand, it still involves an indoor test on athletes that need to familiarize with treadmill running [9] and can hardly perform the same accelerations of a race sprint.

Running specific prostheses (RSP), i.e., curved composite laminates clamped to the amputee’s tibial socket or prosthetic tibial pylon, are fundamental assistive technologies for Paralympic sprint runners and long jumpers. Since their introduction [10], researchers recognized that structural properties of a running prosthetic foot (RPF) are influencing the behavior of athletes due to ground interaction and the effects on the athlete’s feeling, comfort, and overall performance. In addition to the intrinsic stiffness of RPF, the alignment between socket, knee (in transfemoral), and foot has become a key factor for the fitting of the prosthesis, depending on athlete’s anthropometry and running characteristics [11]. Typically, static stiffness of RPFs has been studied on uniaxial test machines equipped with upper clamps for the support of the RPF and introducing fixed or sliding devices for the simulation of the ground floor at the RPF tip, at different ground angles and with different restraint conditions [12,13,14]. Recently, the development of a multicomponent test bench for the dynamic characterization of running specific prostheses allowed the evaluation of static stiffness properties at different ground angles and load combinations, including dynamic drop tests [15].

Given the importance of knowing the structural loads acting on RPF, to overcome the limitations of the three aforementioned approaches in GRF field collection, a fourth approach was adopted in the present work. Reversing the usual adoption of ground/treadmill-based force platforms, a set of wearable sensors was introduced on the prosthetic leg. Two commercial RPF, a J-shaped and a C-shaped, were instrumented with a set of strain gauge bridges and calibrated statically on the multicomponent test bench [15] to obtain two instrumented running prosthetic foot (iRPF). The two iRPF were connected to a wearable data logger for measuring the structural loads acting at the RPF clamp on the sagittal plane during sprints or long jumps. The two iRPF were validated during dynamic sprints and long jumps against ground based force platforms and subsequently used for the collection of multiple step loads in track tests involving two elite Paralympic athletes.

## 2. Materials and Methods

### 2.1. Sensor Design Specifications

The functional requirements of each instrumented running prosthetic foot were defined to ensure its application as a wearable sensor during real sprint and jump events as follows: (i) maximum number of loads components, (ii) load component measured with the highest degree of decoupling, and (iii) minimum additional weight on the leg (< 0.15 kg) to avoid any disturbance to the athlete.

The iRPF was then developed based on these requirements. The solution of adding a commercial multiaxial load cell at the RPF clamp was avoided due to the usual high mass of such devices, even if miniaturized; recent attempts of developing an instrumented pyramidal joint [16] acting only in the sagittal plane, despite attractive, were considered at this stage not reliable and robust enough for the high-duty function of such RPF during sport events.

On the contrary, based on the observation of the current geometry of the C-shaped or J-shaped feet, it was decided to let each athlete use its own RPF, with its usual clamp and alignment and to apply strain gauge bridges directly to the foot, after adopting appropriate disposition of the bridges and calibration procedures, as reported below.

### 2.2. Terminology and Reference Systems

RSP are assistive technologies used by Paralympic sprint runners and long jumpers to reach their maximal performances. Considering athletes with a transtibial amputation (TT), the RPF is directly clamped to the socket; in the case of transfemoral amputees (TF) or knee-disarticulated athletes (KD), the RPF is connected via a link (or pylon) to the knee that, in turn, is connected to the socket. RPF can typically be J shaped or C shaped; loads acting on the RSP come from ground reaction forces (GRF) and reach the leg through a complex combination of loads parallel and normal to the prosthetic link/socket assembly.

As a first step, a common set of terms was introduced to enable the possibility of expressing loads acting on the RPF of a TT athlete either in the Global (Figure 1a), in the Foot (Figure 1b), or in the Socket (Figure 1c) reference systems. To do so, clear definitions of axes alignment with respect to the volumetric shape of each component of the prosthetic system are needed. In general, multiple reference systems can be applied to the same component, in particular to the RPF where, for instance, the Foot Clamp (Figure 1b,c) system plays an important role in the adjustment of socket/link/foot alignment. Finally, loads measured in the Foot reference system (Figure 2a) can also be expressed in the Shank or in the Thigh (Figure 2) reference systems, centered at the Knee Joint Center (KJC) or at the Hip Joint Center (HJC), respectively, provided that synchronous motion capture techniques with sufficient accuracy are adopted. The detailed definition of the adopted reference systems is reported in Appendix A.

Loads measured by a force platform are typically expressed in the Global Reference System (GRS), where *X*_G_ is horizontal and *Y*_G_ is vertical. In our work, a Foot Reference System (FRS) was embedded on the instrumented RPF to resolve the GRF accordingly (Figure 3): its orientation relative to the socket (TT amputees) or to the link (TF and KD) can change only due to alignment adjustments that prosthetists can apply, e.g., angle γ in Figure 1c. The Socket Reference System (SRS) is, in turn, referable to body anatomical segments like the Shank and the Thigh if residual limb relative motion is taken into account.

### 2.3. Sensing Concept of the Instrumented Running Prosthetic Foot (iRPF)

Two strain gauge pure bending bridges, BRG1 and BRG2, were applied to the proximal straight portion of the upper stiffer part of the foot, with a span **a** in between (Figure 4). The two bridges mid locations identify a straight line pointing forward that was identified as the Foot *x*-axis, named “*x*_F_.” The axis perpendicular to *x*_F_ is then *y*_F_, laying in the foot plane: together, they identify the foot system of reference, centered at the first bridge BRG1, named “point F.” A third pure bending bridge BRG3 was placed on the curved portion of the blade (Figure 4) in correspondence of the tangency point between the midline of the RPF bent and a line perpendicular to axis *x*_F_. The *x* and *y* distances of BRG3 from the origin F of the Foot reference system were named “**b**” and “**c,**” respectively.

The position adopted for the three bridges ideally allows to decouple the effect of loads *Fx*_F_ and *Fy*_F_, that are, respectively, along the *x*_F_ axis and the *y*_F_ axis. This implies the possibility of resolving the three load components *Fx***_F_**, *Fy*_F_, and *Mz*_F_ acting in the Foot plane at point F from the measure of the three bending moments *Mz*_1_, *Mz*_2_, and *Mz*_3_ at the three bridges BRG1, BRG2, and BRG3 [17]. In fact, given the two bending moments *Mz*_1_ and *Mz*_2_, load *Fy*_F_ corresponds to a shear load in the beam portion between sections 1 and 2; therefore, it is proportional to the difference between *Mz*_2_ and *Mz*_1_. On the other hand, the bending moment *Mz*_3_ is due not only to the effect of *Mz*_1_ and *Fy*_F_ with lever arm **b** but also to *Fx*_F_ with lever arm **c**. Knowing *Mz*_1_ and *Fy*_F_, the missing bending contribution to *Mz*_3_ is due to *Fx*_F_ that can therefore be obtained by difference.

### 2.4. Development of the Instrumented Running Prosthetic Foot (iRPF)

Two RPF were considered. The first was an *Össur Cheetah Xtreme* Cat. 5 foot, the second was an *Ottobock, 1E91 Runner, Standard* Cat. 4. Both feet were brand new feet and were adopted as they are chosen by the two elite athletes involved in the field tests.

*Ossur* J-shaped foot presents a thick proximal straight portion (thickness tapering from 15 mm to 13 m) that is usually clamped to the socket of a TT athlete or to the bracket of a KD/TF athlete. The foot presents also a stiff straight portion right outside the clamp that is pointing backwards/downwards and that was identified as a beam section suitable for hosting bridges BRG1 and BRG2 (Figure 5a). BRG3 was placed in correspondence of the bent portion of the blade.

The *Ottobock* C-shaped foot has a special curved proximal extremity with an inner buttonhole where the clamp can be positioned between ticks labelled −4° to +4°, corresponding to a dorsiflexed or a plantarflexed foot orientation, respectively. This slider allows changing the foot orientation while keeping the clamp axis steadily pointed to a reference red label placed in proximity of the foot distal tip, as it can be noticed in Figure 5b. From the foot clamp area, which is usually connected to the socket of a TT athlete or to the tibia link of a KD/TF athlete, the foot presents a short straight portion of constant thickness 11 mm which is pointing backwards/downwards and that again can be identified as a beam section suitable for hosting BRG1 and BRG2.

The three bending bridges were strain gauged on each foot using Kyowa strain gauges type KFG-3–120-C1–23L3M3R. At each bridge, two strain gauges were glued to the two opposite faces of the RPF surface and connected as half bridge to a cable ending with a 9 pin D plug, similarly to other consolidated experiences [17]. The leadwires were deployed along the side wall of the RPF and protected with a thin layer of silicon rubber as shown in Figure 5.

### 2.5. Static Calibration of the Instrumented Running Prosthetic Foot (iRPF)

The calibration of the iRPF was performed on a multicomponent test bench developed for the project and described in [15]; the iRPF were clamped at a known position on a 6 axis load cell (ME, K6D80/2 kN/100 Nm) as shown in Figure 6. The Load cell vertical axis *y*_C_ was aligned with the vertical actuator of the bench. The tip of the RPF was in contact with a horizontal slider equipped with a tartan surface and a 3-axis load cell (ME, K3D120 ± 5 kN/VA). Loads were applied in *y*_C_ direction by a servohydraulic MTS 242 actuator of 15 kN range, up to a maximum *Fy*_C_ value of 1500 N: a second servohydraulic MTS 242 actuator was controlled in *x*_C_ direction in order to ensure either zero force values, zero horizontal displacement, or the desired force values for validation tests (Figure 6a). Bending moments applied to the three bridges at each load step were measured by the combination of three sagittal load components measured at the load cell and the instantaneous location of bridges expressed by the *x*–*y* distances from the load cell origin, as shown in Figure 6b. This was performed at all load levels, therefore, including the large displacements implied in the RPF deflections. Output from Load cell and bridges were synchronously measured with a portable data logger Somat EDAQ connected to the bridges.

Formulas adopted for the calculation of bending moment acting at each bridge location are reported in (1).
(1)$$Mz_{1}=Mz_{C}+Fy_{C}\cdot x_{1}-Fx_{C}\cdot y_{1}\;Mz_{2}=Mz_{C}+Fy_{C}\cdot x_{2}-Fx_{C}\cdot y_{2}\;Mz_{3}=Mz_{C}+Fy_{C}\cdot x_{3}-Fx_{C}\cdot y_{3}$$

The calibration procedure allowed evaluating the response of the three bending strain gauge bridge channels to known bending moments and calculating the linear calibration constant K of each bridge, as expressed by the following Equation (2):(2)$$Mz_{1}\left[Nm\right]=K_{1}\left[\frac{Nm}{\frac{mV}{V}}\right]*brg_{1}\;\left[\frac{mV}{V}\right]\;Mz_{2}\left[Nm\right]=K_{2}\left[\frac{Nm}{\frac{mV}{V}}\right]*brg_{2}\;\left[\frac{mV}{V}\right]\;Mz_{3}\left[Nm\right]=K_{3}\left[\frac{Nm}{\frac{mV}{V}}\right]*brg_{3}\;\left[\frac{mV}{V}\right]$$
where *Mz*_1_, *Mz*_2_, and *Mz*_3_ are the bending moment applied at the three bridges, *brg*_1_, *brg*_2_, and *brg*_3_ are the three-signal output from the bridge channels, and K_1_, K_2_, and K_3_ are the three calibration constants of the bridges.

Bending moments *Mz_i_* sensed at the three bridges can be in turn combined with geometrical distances **a, b,** and **c** characteristics of the bridge disposition represented in Figure 4 to give the loads acting on the Foot at the Reference System Origin, named “F” and located on BRG1. Loads at the Foot reference point F can be obtained by following Equations (3):(3)$$Fx_{F}\;\left[N\right]=\frac{a-b}{a\cdot c}\cdot Mz_{1}+\frac{b}{a\cdot c}\cdot Mz_{2}-\frac{Mz_{3}}{c}\;Fy_{F}\;\left[N\right]=\frac{Mz_{2}-Mz_{1}}{a}\;Mz_{F}\;\left[Nmm\right]=Mz_{1}$$

The knowledge of loads at the Foot Reference Point F together with the Foot absolute orientation angle θ_FO_ with respect to the Global reference system allows the calculation of ground reaction forces GRFx and GRFy in the Global reference system, as described in Figure 3 and Figure 7.

In addition, the position of the foot with respect to the socket or the link after a specific alignment adopted by the prosthetist is known and corresponds to a rigid rototranslation. Loads measured in the Foot reference system can be easily resolved in the Clamp reference system (introducing the known angle δ of Figure 1b) or the Socket reference system (introducing the known angles δ + γ) in the case of a TT amputee, as well as in the Link reference system or the Socket reference system in the case of a KD or TF amputee with fully extended knee. The knowledge of loads acting in the Shank or Thigh reference systems requires the knowledge of instantaneous limb orientation as well as the assumption of negligible relative motion between stomp and socket.

Equations (1)–(3) were introduced to measure the loads in the Foot reference system; after that, static validation tests were performed on the two iRPF by applying combined loads in the *x*_C_ and *y*_C_ direction as reported in Figure 6b. *Fy*_C_ and *Fx*_C_ loads were measured by the 6-axis load cell while the iRPF was loaded by the vertical cylinder in the *y*_C_ direction, with the horizontal cylinder set to null displacement. This horizontal restraint corresponded to the FDE configuration introduced by Dyer et al. [12], i.e., preventing the foot tip from advancing in *x*_C_ direction, a negative *Fx*_C_ force was generated and resulted applied together with *Fy*_C_. Maximum values of *Fy*_C_ and *Fx*_C_ applied forces were 1352 and −661 N for *Ossur* iRPF and 1489 and −559 N for *Ottobock* iRPF, respectively. The maximum percent error at the peak load was calculated as the deviation between the load predicted by the calibrated iRPF and the value measured by the load cell.

### 2.6. Dynamic Validation of the Instrumented Running Prosthetic Foot (iRPF) during Field Tests

Dynamic validation tests were carried out in field on an instrumented, outdoor athletic track (Budrio, Italy); the track was equipped with a Kistler 600 mm × 400 mm force platform (sampling rate 5 kHz, covered with tartan), embedded at 30 m from the start of the long jump run-in track and at the takeoff board. Sprint and running events were then performed with athletes from the Italian national team. Together with the force platform, two Sony high-speed cameras Alpha RX-II (at 1000 Hz) were used to capture the steps over the platforms, and the Optogait system (Microgate, Italy) was used in the tests to capture spatiotemporal parameters of running (Figure 8). Structural data were collected on a GET M40 portable data logger from Athena (Longare, IT) at 500 Hz per channel, placed into a small backpack with a total added mass of 2.3 kg.

An elite male athlete with unilateral right TT amputation (TT1, height 1.72 m, mass 67 kg) performed several sprint tests using the instrumented *Ossur Cheetah Xtreme* Cat. 5 on the track equipped with the force platform, after signing an informed consent. Out of 15 runs, 5 runs showed successful clean steps over the platform that were collected both by the Kistler systems and the portable Somat Datalogger (Figure 8b).

An Olympic medalist female athlete with unilateral left KD amputation (KD2, height 1.66 m, mass 58 kg) performed also several sprint tests using the instrumented *Ottobock Runner* Cat. 4 on the track equipped with the force platform, after signing an informed consent. Out of 14 runs, six steps were successfully collected simultaneously by the force platform and the wearable sensory system (Figure 8c).

### 2.7. Track Sprint and Long Jump Data Collection Using the Instrumented Running Prosthetic Foot (iRPF)

A second session of field tests was carried out on the same instrumented outdoor athletic track (Budrio, Italy) (Figure 9); the Kistler force platform was used again in the long jump run-in track at the takeoff board (Figure 9b). Sprint and jumping events were then performed by athlete KD2. A set of free sprints was also performed by the athlete in the straight portion of the free track that was not equipped with force platform but only by Optogait system (Microgate, Italy) (Figure 9c).

In this case, the Xsens Moven suit (Xsens, Enschede, The Netherlands) was worn by the athlete to capture the absolute orientation of the foot during stance step by step with a fully wearable solution; the application to an amputee runner required some adaptations. With athlete KD2, following the denomination of Xsens Moven inertial motion unit (IMU) sensors, the Upper LEG Left sensor was placed on the THIGH, the Lower LEG Left sensor on the FOOT proximal End, and the FOOT Left sensor was applied to Left FOOT distal End (Figure 9a). This implied that the Lower LEG Left sensor described the orientation of the iRPF and, interestingly, the (virtual) Left Ankle Joint collected with the Moven suit represented the sagittal angular deflection of iRPF under realistic running conditions. Structural data were collected on a GET M40 portable data logger at 500 Hz per channel, whereas the Xsens inertial suit collected samples at 240 Hz.

## 3. Results

### 3.1. Calibration Tests

Results of calibration tests showed that all bridges installed on the instrumented feet responded linearly to the application of bending moments. The sensitivity curves of three bridges of *Ossur* iRPF are reported in Figure 10a, whereas sensitivity curves of *Ottobock* iRPF bridges are presented in Figure 10b. Geometrical dimensions of bridge disposition (as described in Figure 4) for the two instrumented RPF are collected in Table 1; Table 1 also reports the calibration constants K of each bridge of the two iRPF, as expressed by Equation (2). Together with each value of K, the R^2^ coefficient of a linear regression of each load ramp are reported.

During static validation tests, the two peak load components *Fx*_C_ and *Fy*_C_ could be estimated with an error of 8.35 % in *x*_C_ and 4.78% in *y*_C_ for the *Ossur* iRPF and of 10.79% in *x*_C_ and 7.75% in *y*_C_ for the *Ottobock* iRPF.

### 3.2. Dynamic Validation Tests

The results of dynamic validation tests of the iRPF allowed comparing the GRFs measured with the Force Platform and those obtained with the iRPF during running tests of athletes TT1 and KD2 on a force platform.

For athlete TT1, running with the *Ossur Cheetah Xtreme* Cat. 5, force components collected in the Clamp reference system from the iRPF (Figure 11a) are combined with the absolute orientation angle of the Clamp reference system obtained with high-speed camera nr. 2 during the step over the force platform (Figure 11b). This allowed resolving the forces from the iRPF in the Global reference system and to plot together the GRF measured by the Kistler force platform at the passage of the athlete (Figure 11c) and those calculated from the iRPF. In Figure 11d, the *X* and *Y* GRF components obtained from the force platform measurement (dotted) and the iRPF loads resolved in the Global reference system (solid) are plotted together, showing a good agreement throughout the stance. Force values were normalized to athlete’s body weight and expressed within the stance duration.

Results of athlete KD2 sprint tests with the *Ottobock Runner* STD Cat. 4 are reported in Figure 12. Force components collected in the Foot reference system from the iRPF (Figure 12a) are now combined with the absolute orientation angle of the Foot reference system obtained with the Xsens Moven suit during the step over the force platform (Figure 12b). GRFs measured in the GRS by the Kistler force platform at athlete’s passage (Figure 12c) were compared to those collected from the iRPF and resolved in the GRS. In Figure 12d, the *X* and *Y* GRF components obtained from force platform measurement (dotted) and from iRPF loads resolved in the Global reference system (solid) are plotted together, showing again good agreement.

Results of athlete KD2 jump tests with the *Ottobock Runner* STD Cat. 4 were analyzed with the same procedure and collected in Figure 13 for two different long jumps. In Figure 13a, the *X* and *Y* GRF components obtained from force platform measurement (dotted) and from iRPF loads resolved in the Global reference system (solid) are plotted together for Jump J21 (jump length 4.50 m), showing encouraging agreement also for such an impulsive event. For comparison, GRF in the GRS for jump J22 (jump length 4.30 m) are presented in Figure 13b.

The maximum peak value errors and the root mean square error (RMSE) normalized to the peak values for the three dynamic validations set of tests, in horizontal *x* and vertical *y* in the GRS directions are collected in Table 2.

### 3.3. Field Tests

Field tests performed on a sprint run by athlete TT1 with the *Ossur* iRPF, together with the validation step recorded on the force platform, allowed to collect multiple steps and to study the progression of sprint loads during the run as shown in Figure 14. Y-Force component curves, expressed in the Foot Clamp reference system, show a progressive shortening of step duration and an increase of peak loads up to step 7. X-Force components show a less regular trend towards an increase of peak loads, with steps 7 and 4 showing the highest peak values.

Two sprint tests performed by athlete KD2 with the *Ottobock* iRPF are compared in Figure 15: Run 10 and Run 12. Thanks to the adoption of the iRPF and the Xsens suit in the two full runs; 12 steps on Run 10 were collected and resolved in the Global reference system over the sprint test, including step 8 over the Kistler force platform. The iRPF allowed the calculation of the horizontal impulse from the affected limb and the comparison between the sprint aiming to the force platform (Run 10, “platform”) and the free sprint on track without platform (Run 12, “track”).

As it can be appreciated from Figure 15, the horizontal impulse measured during all steps of sprint test Run 10 aiming to hit a force platform is predominantly negative, whereas the horizontal impulse measured in a free sprint test on a track without platform aiming to the best run time (Run 12) shows predominantly a propulsive positive impulse.

## 4. Discussion

The present study aimed to develop two instrumented RPF for the collection of structural loads acting during track tests using wearable sensors.

Motivation of the study was the awareness that knowledge of loads acting on RSPs is an essential information; the measure of loads is essential for (i) evaluating the athletes running technique, (ii) designing RPF with appropriate structural properties, and (iii) developing biomechanical models of runners aimed to optimize the sprint or jump performances.

The direct application of strain gauge bridges to the carbon blades transformed the RPF into wearable sensors with the minimum weight effect, null stiffness variations, and no disturbance to the possible foot adjustments that the orthopedic technicians adopt during field tests searching for the optimal performance on the basis of athlete’s feedback and time laps evidences [11]

Results of calibration tests showed the high sensitivity of the strain gauge bridge also to loads that were about half the peak loads recorded in the field: this confirmed that the adopted approach of using the RPF as a wearable sensor provided suitable signals for load reconstruction. In addition, it can be noticed that the sensitivity curves of *brg*1 and *brg*2 obtained from *Ottobock* iRPF do not differ more than 10%, whereas in the *Ossur* iRPF *brg*2 shows a significant higher sensitivity than *brg*1; this can be explained by the fact that the blade portion adopted for *brg*1 and *brg*2 in *Ottobock* foot presented a uniform thickness and width, as well as possibly the same composite laminate. This can be taken as an indirect confirmation of the quality of the strain gauged bridges. The blade portion adopted for *brg*1 and *brg*2 in *Ossur* foot instead did not presented a uniform thickness and width; this led to different values of sensitivity, possibly induced also by differences in the composite laminate that was unknown for both feet.

Loads collected on force platform with elite athletes such as TT1 and KD2, when compared with available data from literature for the affected limb [7], resulted in fair agreement both in terms of curve shape and GRF peak values. From the analysis of Figure 10, Figure 11, Figure 13 and Table 2, the accuracy in the prediction of GRF forces was judged to be good in vertical *y*_G_ direction and acceptable in horizontal *x*_G_ direction. Considering that sprinters preserve a margin of variability in their steps and that a scatter is intrinsic in the GRF plots of elite athletes, the static and dynamic validation of the iRPF were considered acceptable for the collection of track loads with a wearable system.

Results of multiple steps, such as plots reported in Figure 14, even when expressed in the Foot Clamp reference frame, can support coaches, athletes, and technicians in the investigation of performance progression during the sprint or the long jump run-in. Moreover, given the rigid geometric connection between the foot and the socket (in TT amputees) or between the foot and the prosthetic knee (in KD and TF amputees), forces and moments acting at the knee of the athlete can be measured in real time and with accuracy along the full run. This will ensure that the knee will work consistently in safe conditions (to avoid knee undesired flexion) and will support coaches and prosthetists in the comparative evaluation of effects that RPF alignment (or full prosthesis) can have on athletes performance, comfort perception, and injury prevention [18,19].

The new approach presented here was successful in measuring not only a single step on a force platform embedded in a track but also in collecting multiple steps in sprint and long jump run-in events; to the best of our knowledge, these results are an original contribution in the field of Paralympic sprint runners [20].

Potentially, the solution allows for the exploration of curve running that is a complex task for amputee sprinters, rarely analyzed. The application of torsional bridge on the iRPF is a future development of this sensory system with implications in the reproduction of measured torsional loads on modified test benches [15].

The main limitation of our approach is that only the affected limbs can be studied, whereas using force platforms or instrumented treadmills both sides are measured. On the other hand, the present approach allows for realistic sprint events in outdoor track or race environment, without introducing the simulated running condition of motorized treadmill or the ambient conditioning of indoor tests. A second limitation is that sensors and calibrations had to be applied and performed on all feet used by the athletes, without the possibility of exchanging rapidly several feet for real-time comparison. Furthermore, only sagittal plane loads are collected so far at the foot clamp.

Using the iRPF integrated with inertial units as shown in the present study resulted to be a powerful tool for the detailed step by step analysis of running style, running performance, and eventually prosthetic alignment effects. The extension to curve running will open new perspective of studies for the improvement of long sprint competitions in terms of athlete’s performance, health, and safety.

## 5. Conclusions

All steps of track sprints and long jumps of two elite athletes were collected thanks to the application of strain sensors on two commercial RPF, their accurate calibration, their static and dynamic validation against force platforms GRF, and integration with an inertial motion capture suit. The two RPF, a J-shaped *Ossur Cheetah Xtreme* Cat. 5 and a C-shaped *Ottobock 1E91 Runner Std.* Cat. 4, were instrumented with three strain gauge bending bridges. The calibration was performed on an innovative multichannel test bench and equations for the prediction of loads acting at the foot clamp or on the socket were introduced.

Dynamic validation tests were performed involving two Olympic ranking athletes, a male TT and a female KD athlete, running on a track equipped with time gates, a force platform and high-speed cameras in the sagittal plane. The comparison between the ground reaction forces measured at the platform and those predicted by the instrumented RPF gave RMSE in sprinting not larger than 12.4% in horizontal and 3.5% in vertical direction. The same validation tests of long jumping takeoff ground reaction forces gave RMSE not greater than 13.4% in horizontal and 7.6% in vertical direction.

The iRPF approach was used to collect track loads over all the steps of realistic sprint tests performed by the athletes, leading to investigations regarding the progression of foot clamp loads along the run and the comparison of braking and propulsive impulse along the full run for the comparative evaluation of running techniques.

## Figures and Tables

**Figure 1 sensors-20-05758-f001:**
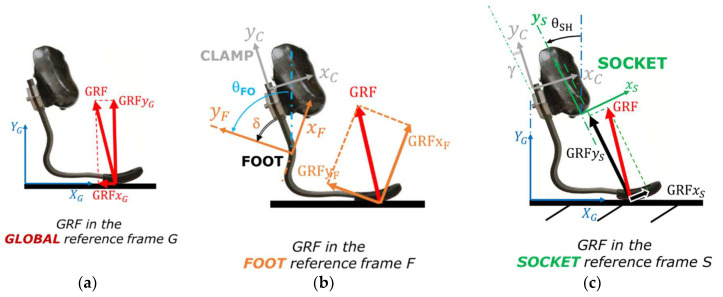
Reference Systems adopted to express the ground reaction forces (GRF) components for a transtibial amputation (TT) running specific prostheses (RSP). (**a**) Global reference system and GRF components, (**b**) Foot and Clamp reference systems with GRF components in the Foot reference system, and (**c**) Socket reference system and GRF components.

**Figure 2 sensors-20-05758-f002:**
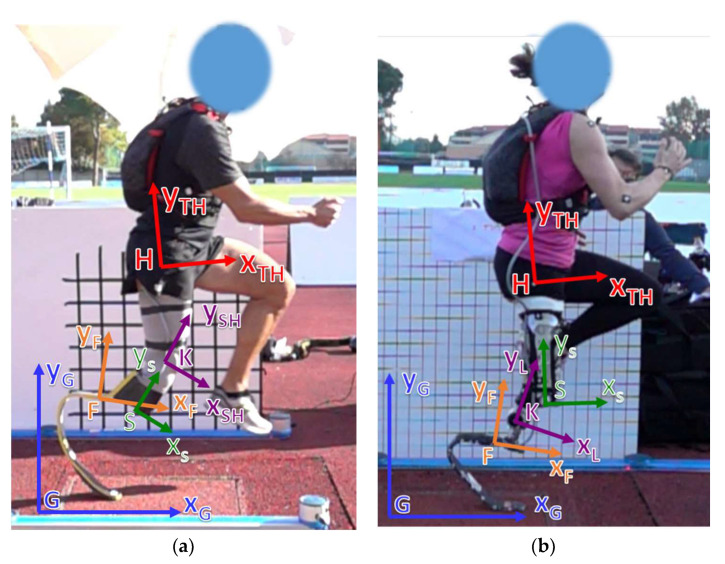
Reference Systems in the sagittal plane associated to athlete’s limbs and to different components of the Running Prosthetic System allow expressing their absolute and relative orientation and resolving loads coming from the ground contact for a (**a**) TT amputee or (**b**) a transfemoral amputees (TF) and knee-disarticulated athletes (KD) amputee. See Appendix A for reference systems definitions.

**Figure 3 sensors-20-05758-f003:**
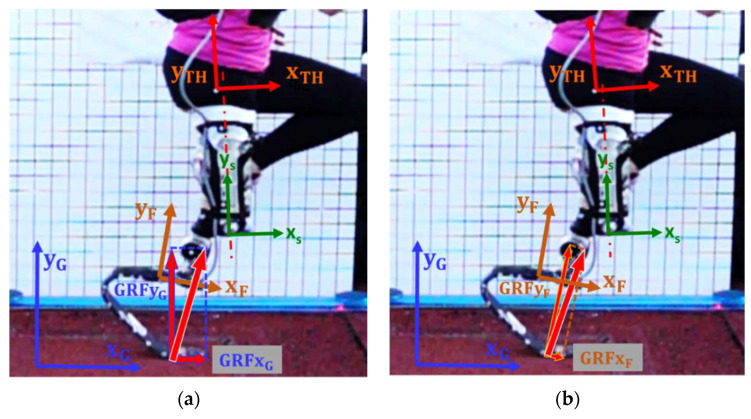
Example of GRF components in the push phase of a KD RSP resolved in different Reference Systems: (**a**) GRF components the Global reference system and (**b**) GRF components in the Foot reference system. See Appendix A for reference systems definitions.

**Figure 4 sensors-20-05758-f004:**
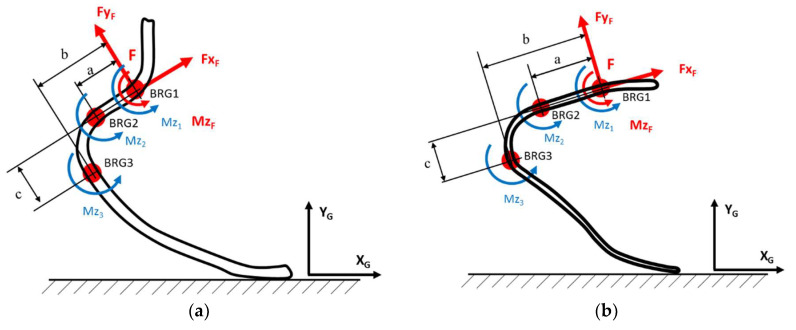
Sensing concept adopted for the instrumented running prosthetic foot. Three bending bridges sensing the bending moment *Mz*_1_, *Mz*_2_, and *Mz*_3_ at known locations allow reconstructing the three load components *Fx*_F_, *Fy*_F_, and *Mz*_F_ at the Foot reference system origin F. (**a**) Example of a J-shaped instrumented running prosthetic feet (RPF). (**b**) Example of a C-shaped instrumented RPF.

**Figure 5 sensors-20-05758-f005:**
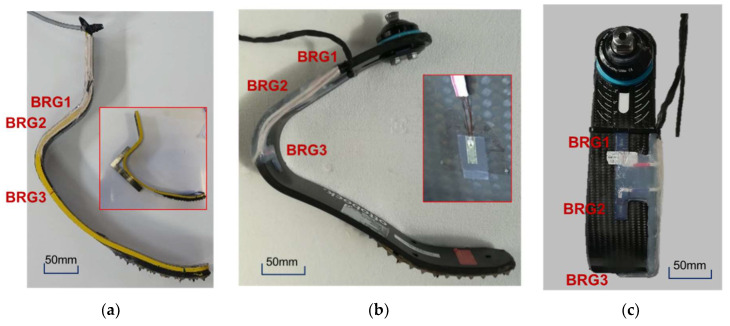
Development of the instrumented RPF: (**a**) strain gauge bridges applied to the J-shaped *Cheetah Xtreme* Cat. 5 RPF, with a detailed example of the relative position between bridge nr. 3 and bridges nr. 1 and 2; (**b**) side view of *Ottobock Runner* Cat. 4 RPF, with a detailed view of the applied strain gauge over the carbon fiber laminate; and (**c**) strain gauge bridges applied to C-shaped *Ottobock Runner* Cat. 4 RPF.

**Figure 6 sensors-20-05758-f006:**
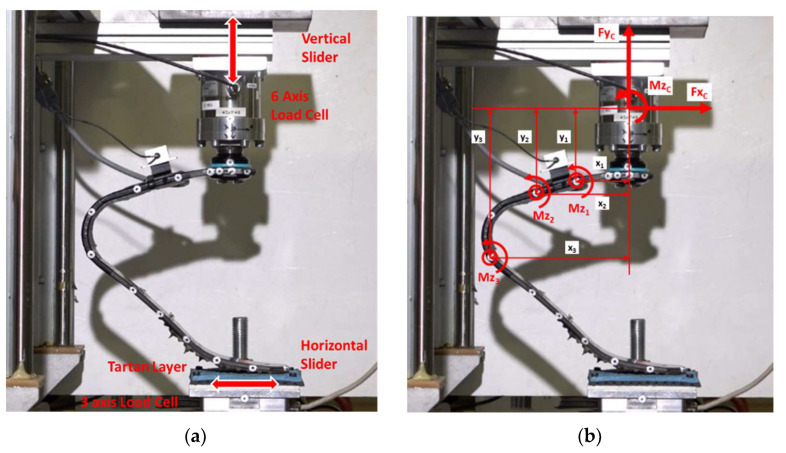
Calibration method of the instrumented RPF: (**a**) the instrumented running prosthetic foot (iRPF) is clamped to a 6-axis load cell and loaded from a controlled vertical slider against a controlled horizontal slider equipped with a 3-axis load cell covered with tartan and (**b**) moments *Mz_i_* applied to the three bending bridges are calculated from the combination of load cell loads and the instantaneous location of the bridges expressed by the coordinates *x* and *y* of each bridge in the load cell system of reference.

**Figure 7 sensors-20-05758-f007:**
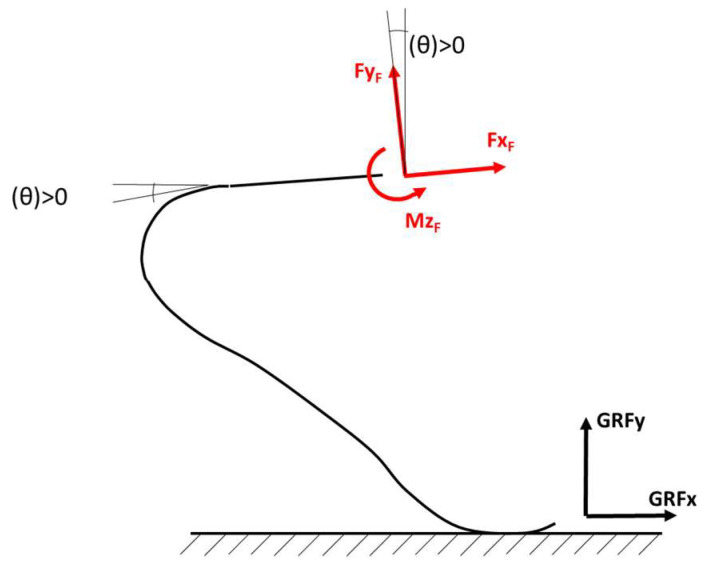
Absolute orientation of the iRPF used for the reconstruction of loads applied to the iRPF from the ground in the Global Reference System.

**Figure 8 sensors-20-05758-f008:**
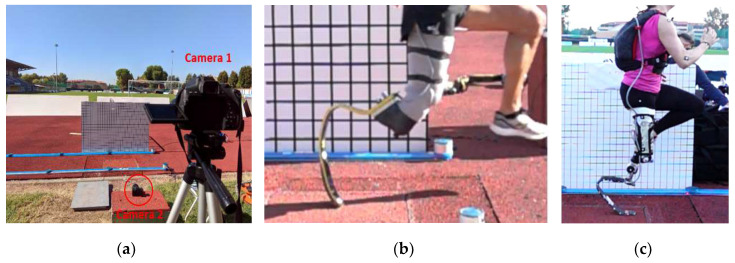
Field test session setup: (**a**) experimental setup on the track equipped with force platform and high-speed video cameras, (**b**) athlete TT1 valid step on force platform with simultaneous collection of GRF from force platform and the iRPF, and (**c**) athlete KD2 valid step on force platform (mirrored image).

**Figure 9 sensors-20-05758-f009:**
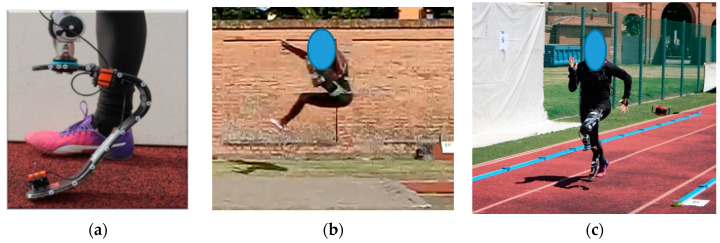
Second field test session setup: (**a**) inertial suit from Xsens applied to athlete KD2, with detail of the RPF arrangement of inertial motion unit (IMU) sensors; (**b**) jumping validation tests with simultaneous collection of GRF from a force platform at the takeoff board and from the iRPF; and (**c**) sprint tests on free track without force platforms.

**Figure 10 sensors-20-05758-f010:**
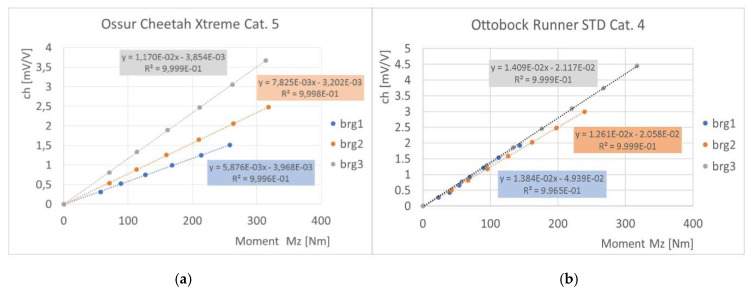
Results of bridge calibration of the instrumented RPF: bridge output (in mV/V) of the three bending bridge channels (brg1, brg2, and brg3) subjected to increasing bending moments; (**a**) *Össur, Cheetah Xtreme* Cat. 5; (**b**) *Ottobock, 1E91 Runner, Standard* Cat. 4.

**Figure 11 sensors-20-05758-f011:**
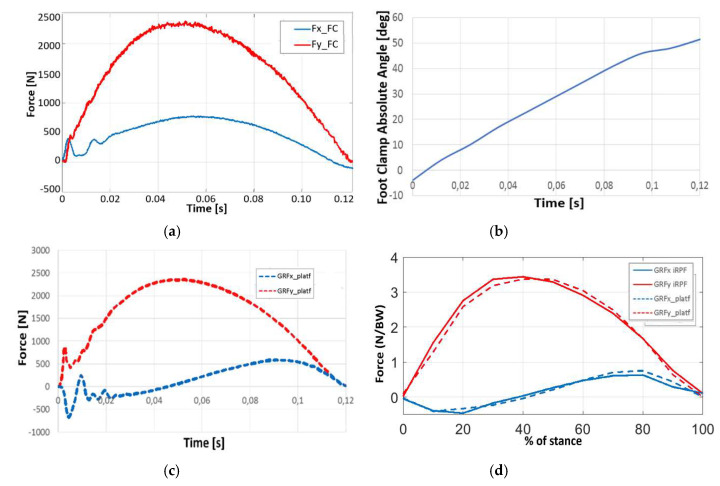
Validation of the Ossur iRPF, by comparison of GRF measured with the Force Platform and with the iRPF during a running test of athlete TT1: (**a**) force components in the Foot Clamp reference system, collected from the iRPF; (**b**) absolute orientation angle of the Foot Clamp reference system obtained by the high-speed camera during the step over the force platform; (**c**) GRF measured by the Kistler force platform at the passage of the athlete; and (**d**) comparison of *X* and *Y* GRF components between the force platform measurement (dotted) and the iRPF loads (solid) resolved in the Global Reference System (GRS) normalized to the stance interval and to athlete’s weight.

**Figure 12 sensors-20-05758-f012:**
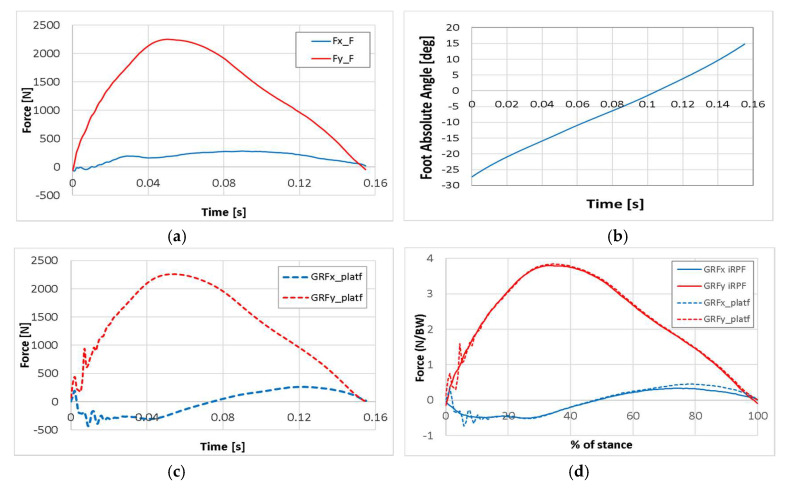
Validation of the Ottobock iRPF, by comparison of GRF measured with the Force Platform and with the iRPF during a running test of athlete KD2: (**a**) force components in the Foot reference system, collected from the iRPF; (**b**) absolute orientation angle of the Foot reference system obtained by the Xsens Moven suit during the step over the force platform; (**c**) GRF measured by the Kistler force platform at the passage of the athlete; and (**d**) comparison of X and Y GRF components between the force platform measurement (dotted) and the iRPF loads resolved in the Global reference system.

**Figure 13 sensors-20-05758-f013:**
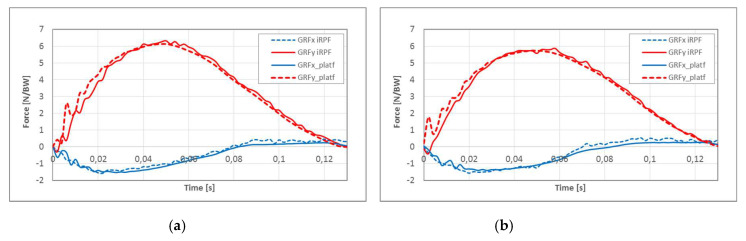
Validation of the Ottobock iRPF, by comparison of GRF measured with the Force Platform and with the iRPF during a long jump test of athlete KD2 loads resolved in the Global reference system: (**a**) validation graph of long jump *n*° 21 (speed at the takeoff = 6.45 m/s, length = 4.50 m) and (**b**) validation graph of long jump *n*° 22 (speed at the takeoff = 6,.45 m/s, length = 4.30 m).

**Figure 14 sensors-20-05758-f014:**
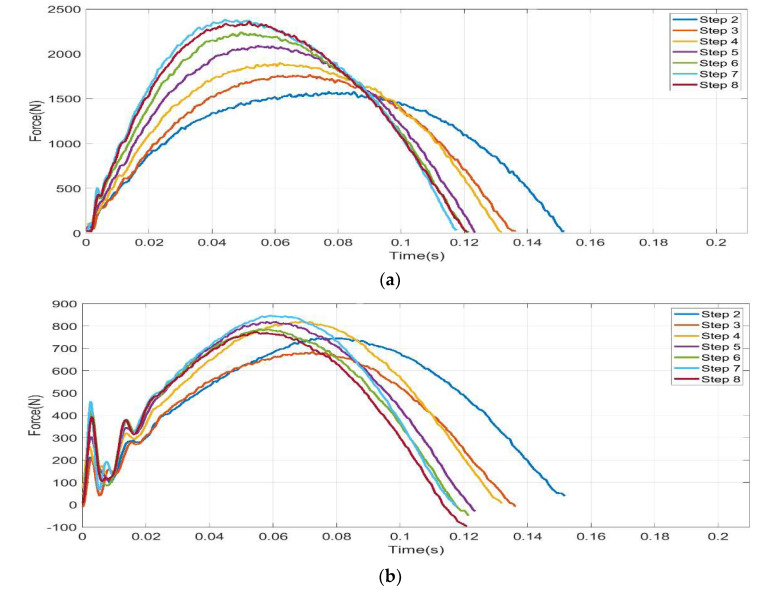
The application of iRPF for multiple step GRF collection with TT1: in the Foot Clamp Reference system, right steps from 2 to 8 are compared during acceleration phase; (**a**) Y Force component and (**b**) X Force component.

**Figure 15 sensors-20-05758-f015:**
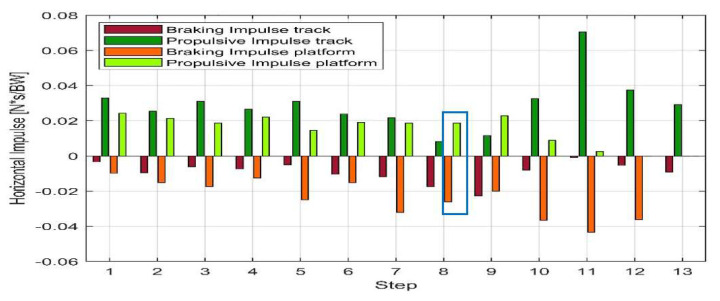
Comparison of the horizontal impulse from the affected limb calculated with the iRPF between the sprint aiming to the force platform and a free sprint on track: all steps of the test aiming to a force platform (Run 10 “platform”) in light colors (platform is on Step 8) and all steps of a free sprint test on track without platform (Run 12, “track”) in dark colors.

**Table 1 sensors-20-05758-t001:** Bridge geometrical parameters (ref. Figure 4) and bridge calibration constants (ref. Equation (2)) resulting for the two instrumented running prosthetic foot (iRPF) developed in the study.

Foot	a (mm)	b (mm)	c (mm)	K1 (Nm·V/mV)	K2 (Nm·V/mV)	K3 (Nm·V/mV)
Ossur Cheetah Xtreme Cat. 5	56	128	93	170,183 (R^2^ = 0.9999)	127,795 (R^2^ = 0.9998)	85,470 (R^2^ = 0.9996)
Ottobock, 1E91 Runner Std Cat. 4	68	141	60	72,254 (R^2^ = 0.9965)	79,302 (R^2^ = 0.9999)	70,972 (R^2^ = 0.9999)

**Table 2 sensors-20-05758-t002:** Maximum peak value errors and root mean square error (RMSE) error for the three dynamic validations set of tests, in horizontal *x* and vertical *y* directions of the Global reference system.

Test	Peak *Fx* (%)	Peak *Fy* (%)	RMSE *Fx* (%)	RMSE *Fy* (%)
Sprint, TT1	−8.2	6.4	7.4	3.5
Sprint, KD2	−12.4	4.3	12.2	2.5
Jump, KD2	6.3	8.2	13.4	7.6

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
