# Peer review of "Development of Instrumented Running Prosthetic Feet for the Collection of Track Loads on Elite Athletes†"

_sensors, 2020, doi:10.3390/s20205758_

Round 1

Reviewer 1 Report

The authors present a meaningful study about the performance of running prosthetic feet. However, there are too many abbreviations of terminologies and measuring parameters (ca. 15) scattered in the main text; The overuse of them really caused the difficulty for me, as a reader, to follow the story line. Therefore, this manuscript needs many improvements before further consideration.

<1> The authors are highly recommended to summarize the experiment and analysis steps into a scheme/diagram. Besides, the reviewer suggests the authors to italicize the names of brands, gears, equipment, etc. to distinguish them among a bundle of abbreviations.

<2> In Figure 5, the photograph of the strain sensor is not clear on the black background. Please modify it. Also, the scale bar is required for optical images.

<3> In Figure 4a, the area below the middle of the foot (ground) and BGR3 has high strain under loading, possibly more than BRG2 and BRG3, because of the thinness and the curvature at that area. Why did the authors not consider the strain gauge there? Is it out of the tracking board?

<4> The pixel on the tracking board (sagittal plane) is quite large, resulting in a high error of x, y, b, and c parameters. The sum of errors in all directions of force > 15% as shown in Table 2 and Conclusion. It means the significance of the data set is < 85%. Do the authors have any solution for it?

<5> In Figures 11-13, why there are negative values of force at the beginning of the plots, while none in Figure 14?    

<6> Some abbreviation has not defined or unified, such as GRS, RMSE error (RMS error?), BRG vs brg, and xF (big F) vs xF (F subscript).

<7> In the reviewer opinion, the title is quite confined if mentioning “elite” athletes. It seemingly limits the applications of the developed iRPF. The authors are suggested to change it.

Author Response

ANSWER TO REVIEWERS

We thank the Editor and the three Reviewer for the time spent in revising the paper and the opportunity of improving the clarity and quality of the paper.

We did our best to understand all comments and implement all requests: we answered to all comments and we hope the set of answers shall satisfy all Reviewers.  

Thanks

REviewer 1

The authors present a meaningful study about the performance of running prosthetic feet. However, there are too many abbreviations of terminologies and measuring parameters (ca. 15) scattered in the main text; The overuse of them really caused the difficulty for me, as a reader, to follow the story line. Therefore, this manuscript needs many improvements before further consideration.

ANSWER: We thank Reviewer #1 for the time spent and the effort put into the revision of the paper. We payed full attention towards the simplification of reading and the explanation/reduction of abbreviations. We added a list of Abbreviations at the end of the paper as Appendix A. We hope the paper improved in clarity and easiness of reading.

<1> The authors are highly recommended to summarize the experiment and analysis steps into a scheme/diagram. Besides, the reviewer suggests the authors to italicize the names of brands, gears, equipment, etc. to distinguish them among a bundle of abbreviations.

ANSWER: Thanks to Reviewer #1 for the suggestions. At this stage, rather than preparing an additional schematic diagram of the experiments, we preferred to reword the final paragraph of the introduction and to use the detailed subtitles of the METHODS sections to guide the reader along the development of experiments. The new version of the final paragraph of the Introduction is as follows:

“…Given the importance of knowing the structural loads acting on RPF, to overcome the limitations of the three aforementioned approaches in GRF field collection, a fourth approach was adopted in the present work by reversing the usual adoption of ground/ treadmill based force platforms and by introducing a set of wearable sensors on the prosthetic leg. Two commercial RPF, a J-shaped and a C-shaped, were instrumented with a set of strain gauge bridges and calibrated statically on the multi-component test bench [15] to obtain two instrumented Running Prosthetic Foot (iRPF). The two iRPF were connected to a wearable data logger for measuring the structural loads acting at the RPF clamp on the sagittal plane during sprints or long jumps, acting as wearable sensors. The two iRPF were validated during dynamic sprint and long jumps events against ground based force platforms and subsequently used for the collection of multiple step loads in track tests involving two elite Paralympic athletes. “

Further, we italicized the brand names as requested and summarized the abbreviations in a List of abbreviations that we reported in Appendix A.

<2> In Figure 5, the photograph of the strain sensor is not clear on the black background. Please modify it. Also, the scale bar is required for optical images.

ANSWER: Thanks to Reviewer #1 for the suggestions. We clarified the photograph and introduced a scale bar.

<3> In Figure 4a, the area below the middle of the foot (ground) and BGR3 has high strain under loading, possibly more than BRG2 and BRG3, because of the thinness and the curvature at that area. Why did the authors not consider the strain gauge there? Is it out of the tracking board?

ANSWER: Thanks to Reviewer #1 for the question that allows us to give a wide explanation. As Reviewer #1 says, the portion of the blade below Bridge 3 is undergoing high strain during the stance of the foot in running. In fact, pilot experiences at the early stages of the project involved the placement of strain bridges in the most distal portion of the foot. However, the first reason why bridges were not applied there was that at high speed and high intensity running the angle of attack of the foot to ground can reach such high negative values that the ground contact comes very close to the location of Bridge3, causing possible damage to strain gauges and cables. Having said that, we have to highlight that the main reason was that it was not easy to decouple the action of Force FyF from FxF. This was achieved only with the disposition described in the present paper that we believe is the original contribution of the work.

<4> The pixel on the tracking board (sagittal plane) is quite large, resulting in a high error of x, y, b, and c parameters. The sum of errors in all directions of force > 15% as shown in Table 2 and Conclusion. It means the significance of the data set is < 85%. Do the authors have any solution for it?

ANSWER: we are not sure about the reference to the “tracking board” by Reviewer #1, but we can try to answer. If reference is given to Figure 4, where three red circles ( pixel?) are associated to the three bridges, then we can clarify that those red circles are just for highlighting the position of the bridges, but the position of bridges, as well as the measure of a, b and c geometrical parameter can be taken with high precision both with mechanical tools and with coordinate measurement machines when available.  If the reference is to the tracking board shown as a background in Figures 2 and 3, we have to say that a,b,c parameters were measured with high accuracy on a reference bench and not estimated by visual analysis.  Regarding the dynamic Force prediction errors in the x and y directions, during sprint as reported in Table 2 and Conclusions, we have to say that the larger contribution to this error (still acceptable in our opinion given the complexity of test conditions and the advantages of the method) has to be found most in the methods adopted for the measure of the absolute foot orientation angle, either with video approach or IMUs approach, rather than in foot calibration precision. Our further investigation is towards more precise sensors of absolute orientation that are insensitive to high impact accelerations like those applied to the foot itself. 

<5> In Figures 11-13, why there are negative values of force at the beginning of the plots, while none in Figure 14?    

ANSWER to Reviewer #1. The different appearance of the Figures mentioned depends on the different reference system adopted for the analysis of Forces and preparing the plot of collected data. This is the reason why we recalled at the very beginning of the paper in Figure 1 the existence of several Reference systems. In Figures 11 and 12 we reported the validation plots of Running tests with J-shaped foot ( Figure 11) and C-shaped foot (Figure 12). On each Figure, negative values appear only in plots expressed in the Global reference system (figure (c) and (d)), because the horizontal FxG force shows initially a negative braking contribution. Forces sensed by the foot, in the FOOT reference system or the FOOT CLAMP reference system, are always positive, also the FxF or FxFC, in Figure 11.a and 12.a. This is also the reference system adopted to plot the consecutive steps of Figure 14: therefore, it is correct that only positive values of FyFC and FxFC are resulting. The plots in Figure 13 are again expressed in the Global Reference System for a long jump event, therefore they show the typical negative portion of Fx that globally initially brakes the athletes after the ground impact, but necessarily becomes positive in the second portion of the stance to gain propulsion. We hope that these comments can clarify things to Reviewer #1.

<6> Some abbreviation has not defined or unified, such as GRS, RMSE error (RMS error?), BRG vs brg, and xF (big F) vs xF (F subscript).

ANSWER: Thanks to Reviewer #1 for highlighting the issues. We introduced missing definitions, harmonized symbols and abbreviations and included a List of Abbreviations. Regarding brg and BRG, we intentionally distinguished between capital BRG, indicating the location of physical sensors, and small “brg”, indicating the channels output. We think that the two different notation remind to each other but do not conflict. Regarding xF or xF (subscripts), we consistently applied the subscript in the text. Apparent differences we believe are coming from the different font adopted in the text (palatino) and in the figures (arial). However, the overall symbols seem not to be misleading.

<7> In the reviewer opinion, the title is quite confined if mentioning “elite” athletes. It seemingly limits the applications of the developed iRPF. The authors are suggested to change it.

ANSWER: Thanks to Reviewer #1 for the interesting suggestion. We agree that the application of iRPF is not limited to the “elite” athletes, and that it is possibly of interest also for growing athletes to improve their technique. We however would like to highlight that the system was developed and validated involving elite athletes, that means under the most stressful conditions.

Reviewer 2 Report

Introduction

Comment 1: The link between the static testing described in the paragraph beginning on line 59 and the experimental evaluation of performance and/or fit is not clear. In the next paragraph the authors mention a multi component test bench for calibration...but a better transition to the contributions presented in the last paragraph would help the reader.

Comment 2: Instrumented wearable Running Prosthetic Foot (iRPF) could be defined in the introduction instead of the Methods on line 79.

Methods

Comment 3: The sentences beginning at 79 should be turned into a paragraph. It is unclear what the authors would like this paragraph to be. It looks like it should be an introduction to the methods, but reads like an outline of the design specifications.

Comment 4: Line 93. Define RSP...

Comment 5: Line 100. The wording is confusing; the ground reaction forces can be represented in any of these coordinate systems. Why would these terms 'distinguish' what system the forces are expressed?

Comment 6: Figures 2 and 3 - It would be helpful to define the variables for the coordinate system in the figure caption. Also, some of the colors chosen for the coordinate axes are difficult to read...particularly the yellow.

Comment 7: 135. The definitions of the axes earlier would be helpful. The authors could also consider moving the equations used to calculate the bending moments to this section to complement the description provided in the text.

Comment 8: Section 2.3 The multiple small blocks of text could be rewritten to form longer paragraphs to improve flow.

Comment 9: The authors could consider reorganizing the methods section. Starting with what feet were selected (was this informed by the subjects that would be tested later?), then introducing how the feet were instrumented, followed by the lab based testing/calibration. Then the authors could continue with the details of the ‘field’ testing of the instrumented feet.

Comment 10: Why was it important to show results from both sprinting and the long jump take off? The authors could better motivate in the introduction why verification of the system for both events was important. The second set of field data with different equipment used to measure kinematics did not flow well.

Results

Comment 11: Line 311 – Turn this into an introductory paragraph or remove the sentences. In fact, work on the sentence and paragraph structure in this entire section.

Comment 12: Figure 10 – This figure could be cleaned up, and do the authors need so many significant figures in their fits?

Comment 13: How many good trials were used in the analysis? In section 2.5 the authors described 4 and 5 good trials used in the analysis. Could all of these trials be shown in the results to give a better idea about variability? Also, why did the authors not use the force plate to collect data with the imus in section  2.6. The authors should provide more details in 2.6 about the experimental protocol. “A set of free sprint” could be clarified.  

Comment 14: All of the figures could be cleaned up and checked for continuity.

Discussion:

Comment 15: Sentence and paragraph structure should be modified throughout this section.

Comment 16: “Motivation of the study was the awareness that knowledge of loads acting on RSPs is an essential information: the measure of loads is essential for (i) evaluating the athletes running technique, (ii) designing RPF with appropriate structural properties, and (iii) developing biomechanical models of runners aimed to optimize the sprint or jump performances.”  It is good to use i and iii to motivate the work, but this paper seems to only cover ii.

Comment 17: Line 469 – now that the authors have a calibrated system that can be used to make measurements of multiple steps on the track it would be a more compelling demonstration of the integrated system’s potential to show the results from hundreds of steps instead of the 7 to 8 steps shown in Figure 14. Did the authors consider having the participants run multiple trials of a set distance? And then looking at steps during different stages of the sprint across the trials? Say, acceleration to top speed or at a steady state speed after the acceleration phase?

Comment 18: Line 476 – Can the authors explain how this would be done? Is there a feature in the participant’s data that should be corrected? Or would these data be considered to be representative of an athlete at ‘peak’ performance?

Author Response

ANSWER TO REVIEWERS

We thank the Editor and the three Reviewer for the time spent in revising the paper and the opportunity of improving the clarity and quality of the paper.

We did our best to understand all comments and implement all requests: we answered to all comments and we hope the set of answers shall satisfy all Reviewers.  

Thanks

Reviewer 2

Introduction

Comment 1: The link between the static testing described in the paragraph beginning on line 59 and the experimental evaluation of performance and/or fit is not clear. In the next paragraph, the authors mention a multi component test bench for calibration...but a better transition to the contributions presented in the last paragraph would help the reader.

ANSWER:     thanks to Reviewer #2 for the suggestion. We reworded the last paragraph and some of the lines before to help the reader’s comprehension. Here is the proposed version:

…..” Given the importance of knowing the structural loads acting on RPF, to overcome the limitations of the three aforementioned approaches in GRF field collection, a fourth approach was adopted in the present work by reversing the usual adoption of ground/ treadmill based force platforms and by introducing a set of wearable sensors on the prosthetic leg. Two commercial RPF, a J-shaped and a C-shaped, were instrumented with a set of strain gauge bridges and calibrated statically on the multi-component test bench [15] to obtain two instrumented Running Prosthetic Foot (iRPF). The two iRPF were prepared for measuring the structural loads acting at the RPF clamp on the sagittal plane during sprints or long jumps, acting as wearable sensors. The two iRPF were validated during dynamic sprint and long jumps events against ground based force platforms and subsequently used for the collection of multiple step loads in track tests involving two elite Paralympic athletes. “

Comment 2: Instrumented wearable Running Prosthetic Foot (iRPF) could be defined in the introduction instead of the Methods on line 79.

ANSWER:     thanks to Reviewer #2 for the suggestion. We defined the iRPF at the end of the Introduction

Methods

Comment 3: The sentences beginning at 79 should be turned into a paragraph. It is unclear what the authors would like this paragraph to be. It looks like it should be an introduction to the methods, but reads like an outline of the design specifications.

ANSWER:     thanks to Reviewer #2 for the suggestion. We inserted a further paragraph titled “Sensor design specifications” and grouped the considerations under it.

Comment 4: Line 93. Define RSP...

ANSWER to Reviewer #2. The acronym RSP stays for Running Specific Prostheses: It has been introduced in the first line of the Abstract (line 15) and in Line 59 of the introduction. As well it has been included in the list of Abbreviations

Comment 5: Line 100. The wording is confusing; the ground reaction forces can be represented in any of these coordinate systems. Why would these terms 'distinguish' what system the forces are expressed?

ANSWER:     thanks to Reviewer #2 for the comment. We modified the sentences in order to avoid confusion. The spirit of the sentence is to highlight that curves, plots and figures of loads acting on the RPF can change considerably if they are expressed in different systems of reference, and researchers and readers shall always mention which is the reference system they adopt in reporting the results. This is evident also in the answer to Comment <5> to Reviewer #1.

Comment 6: Figures 2 and 3 - It would be helpful to define the variables for the coordinate system in the figure caption. Also, some of the colors chosen for the coordinate axes are difficult to read...particularly the yellow.

ANSWER:     we thank Reviewer #2 for the suggestion. We introduced a detailed description of the reference systems in an Appendix of the paper: rather than in the figure caption, due to its length. We modified the colors of axes to improve readability. The Appendix B is here reported for completeness.

Appendix B. Description of Reference systems (Figures 1-3), in the sagittal plane.

Global Reference System (GRS):

Origin: at a corner of the force platform embedded in the ground/treadmill.

XG axis: parallel to the ground, positive in the running direction. 

YG axis: normal to the ground, positive upwards. 

Socket Reference System (SRS):

Origin: at the most distal point of the socket axis.

yS axis: parallel to the socket longitudinal axis, positive upwards. 

xS axis: normal to yS, positive in the anterior direction. 

Clamp Reference System (CRS):

J-shaped foot:

Origin: at the most proximal point of the foot/socket clamp.

yC axis: parallel to the foot proximal straight clamp portion, positive upwards. 

xC axis: normal to yC, positive in the anterior direction. 

C-shaped foot:

Origin: at the most distal point of the foot/pylon clamp.

yC axis: parallel to the pylon longitudinal axis, positive upwards. 

xC axis: normal to yC, positive in the anterior direction. 

Foot Reference System (FRS):

J-shaped foot:

Origin: at the most proximal point of the foot straight portion distal to the clamp.

xF axis: parallel to the foot straight portion distal to the clamp, positive in the anterior direction. 

yF axis: normal to xF, positive upwards. 

C-shaped foot:

Origin: at the most proximal point of the foot straight portion distal to the clamp.

xF axis: parallel to the foot straight portion distal to the clamp, positive in the anterior direction. 

yF axis: normal to xF, positive upwards. 

Shank Reference System (KRS):

For TT amputees:

Origin: at the anatomical Knee Joint Center (KJC).

ySH axis: parallel to the stomp longitudinal axis, positive upwards. 

xSH axis: normal to ySH, positive in the anterior direction

Pylon Reference System (LRS):

For KD and TF amputees:

Origin: at the mechanical Knee Joint Center (KJC).

yL axis: parallel to the pylon longitudinal axis, positive upwards. 

xL axis: normal to yL, positive in the anterior direction

Thigh Reference System (TRS):

Origin: at the anatomical Hip Joint Center (HJC).

yTH axis: parallel to the thigh longitudinal axis, positive upwards. 

xTH axis: normal to yTH, positive in the anterior direction

Comment 7: 135. The definitions of the axes earlier would be helpful. The authors could also consider moving the equations used to calculate the bending moments to this section to complement the description provided in the text.

ANSWER:     thanks to Reviewer #2 for the proposal. This is also expressed in Comment 10. We acknowledge that in Reviewer #2 point of view the flow of sections may be rearranged. Having considered his/her suggestions, we would prefer to keep the section sequence as it is, because rather than to a temporal sequence of actions, we think that the proposed sequence may help the reader to follow the methodological approach. In that sense, the steps were: 1)Reference Terminology, 2) Sensing Concept, 3)Foot Selection and sensor application, 4)Static calibration for load reconstruction, 5)Dynamic validation against force platforms, 6)Track tests with iRPF as wearable system.

Comment 8: Section 2.3 The multiple small blocks of text could be rewritten to form longer paragraphs to improve flow.

ANSWER:     thanks to Reviewer #2 for the proposal. We kept the paragraph subdivision as each paragraph was addressing a different part of information: 1st) overview, 2nd) Ossur J-shaped, 3rd) Ottobock C shaped, 4th) strain sensors application. We hope this explanation may be acceptable.

Comment 9: The authors could consider reorganizing the methods section. Starting with what feet were selected (was this informed by the subjects that would be tested later?), then introducing how the feet were instrumented, followed by the lab based testing/calibration. Then the authors could continue with the details of the ‘field’ testing of the instrumented feet.

ANSWER:     thanks to Reviewer #2 for the proposal. We addressed this request in the previous answer to Comment 7. In addition to that, we can say that the two feet presented in the study were selected as they were the feet adopted by the two elite athletes recruited for the study. This is a peculiar need of such approach: each athlete’s foot need to be instrumented, calibrated and validated. We understand (and mentioned in the discussion) that this is a limitation of the study, for the big effort needed in the preparation of a wide set of instrumented feet.

Comment 10: Why was it important to show results from both sprinting and the long jump take off? The authors could better motivate in the introduction why verification of the system for both events was important. The second set of field data with different equipment used to measure kinematics did not flow well.

ANSWER:     thanks to Reviewer #2 for the question. We addressed both sprint running and long jumping as these two disciplines are typically performed by the same athletes, for instance athlete KD2. In some cases, as KD2, the same foot is used for both disciplines. In other cases, a foot with a higher stiffness category is used for the long jump. Given the high intensity of both disciplines, we tried to validate the method and present the results for each of the two, to give further robustness to the approach and to encourage other researchers to adopt it for both disciplines. Regarding the description of the second kinematic system, we reworded the text.

Results

Comment 11: Line 311 – Turn this into an introductory paragraph or remove the sentences. In fact, work on the sentence and paragraph structure in this entire section.

ANSWER:     thanks to Reviewer #2 for the proposal. We removed the sentence and revised the section

Comment 12: Figure 10 – This figure could be cleaned up, and do the authors need so many significant figures in their fits?

ANSWER:     thanks to Reviewer #2 for the proposal. We are aware that the Figure 10 is quite dense of information, but we preferred to keep the full expression of the sensitivity curves obtained at the bridges after static calibration. The calibration values collected and reported in Table 1 are exactly derived from these sensitivity, and the number of significant figures was needed to avoid amplification of calibration errors.

Comment 13: How many good trials were used in the analysis? In section 2.5 the authors described 4 and 5 good trials used in the analysis. Could all of these trials be shown in the results to give a better idea about variability? Also, why did the authors not use the force plate to collect data with the imus in section  2.6. The authors should provide more details in 2.6 about the experimental protocol. “A set of free sprint” could be clarified.  

ANSWER:     thanks to Reviewer #2 for the question and suggestions. In fact more than one run with successful data collection was recorded both with athlete TT1 and KD2. Given the extension of the work and the complexity of the different tests performed, we did not considered to include the variability of the repeated tests as a result but to focus more on the workflow of the instrumented foot development. Results of multiple steps will be presented in a further publication. Regarding test with IMUs in section 2.6, they were also performed with the force platforms, and results are those presented in Figure 12 for running and 13 for lung jumping: we will try to make this more clear in the text.

Comment 14: All of the figures could be cleaned up and checked for continuity.

ANSWER:     thanks to Reviewer #2 for the proposal. We corrected the figure accordingly

Discussion:

Comment 15: Sentence and paragraph structure should be modified throughout this section.

ANSWER to Reviewer #2. We are not sure about the request of Reviewer #2However we reanalyzed the section in order to improve its form and readability.

Comment 16: “Motivation of the study was the awareness that knowledge of loads acting on RSPs is an essential information: the measure of loads is essential for (i) evaluating the athletes running technique, (ii) designing RPF with appropriate structural properties, and (iii) developing biomechanical models of runners aimed to optimize the sprint or jump performances.”  It is good to use i and iii to motivate the work, but this paper seems to only cover ii.

ANSWER to Reviewer #2. We thank the Reviewer for the comments. In fact, the paper reports an activity that is part of a wider project where aim (i) and (iii) are under development and flow in parallel to this work. Actually, in Ref. [11] the system was already used to analyze the running technique and to search for improvements due to changes in the alignment: this can be considered aim (i). We reworded slightly the paragraph according to the suggestion, in order to underline which aim was more closely addressed and which are in perspective.

Comment 17: Line 469 – now that the authors have a calibrated system that can be used to make measurements of multiple steps on the track it would be a more compelling demonstration of the integrated system’s potential to show the results from hundreds of steps instead of the 7 to 8 steps shown in Figure 14. Did the authors consider having the participants run multiple trials of a set distance? And then looking at steps during different stages of the sprint across the trials? Say, acceleration to top speed or at a steady state speed after the acceleration phase?

ANSWER: we thank Reviewer #2 for the comments and suggestions. It seems that the aim of the work and the advantages of the approach have been exactly interpreted by Reviewer #2. We believe that the promising application of the wearable system is the possibility of collecting several steps along the sprint, and to compare several trials and the development of the loads along the run as suggested. Indeed, the two Figures we propose, Figure 14 and Figure 15 at the end of the paper report, exactly this type of quantity. In particular, Figure 14 reports the data as collected at the Foot Clamp in the acceleration phase, and if analyzed further can give information on how the athlete changes its step duration, intensity and Forward/Vertical ratio during the 16 steps of the run ( as the 8 steps are only of the affected side). Figure 15, despite its apparent simplicity, encloses a complex data analysis as for each step loads collected by the iRPF in the Foot Reference system need to be synchronized with the absolute inclination of the Foot Reference system captured with the inertial suite and resolved in the Global reference system. Then, the horizontal force FxG in the global needs to be analyzed for computing the negative (braking) Impulse and the positive (Propulsive) Impulse, in order to obtain the histogram of Figure 15. This has been done for a free sprint on a track (Run 12) and for a sprint performed to hit the platform (Run 10): the number of successful runs on the platforms, in terms of correct foot placement, has not been so high so far. Having said that, we hope that the idea behind the iRPF can be transmitted with this set of results, as Reviewer #2 got it, and to start from these first set of results for collecting further data to be presented in a future work.

Comment 18: Line 476 – Can the authors explain how this would be done? Is there a feature in the participant’s data that should be corrected? Or would these data be considered to be representative of an athlete at ‘peak’ performance?

ANSWER: we thank Reviewer #2 for the comment and requests of clarification. Indeed we are working on that as research progresses. It is not widely understood which are the determinants of peak performances, in particular for amputees. Some knowledge about abled-bodied athletes is not  immediately transferable to TT amputees and KD or TF amputees. Several groups are doing research on that. So, building experience and collecting data will enable the analysis of intra-subject best performance and inter-subjects best performances. Ideally, from the analysis of wearable sensory data applied to several steps of multiple trials the analysis may guide towards the determination of peak performance indicator. In addition, the advantage of wearable sensors such as iRPF is the possibility of quantifying directly the kinetic quantities acting at the knee or the hip, that are typically the object of a complex motion capture test session involving ground force platforms or instrumented treadmills.

Reviewer 3 Report

Please see the comments in the attached file.

Author Response

ANSWER TO REVIEWERS

We thank the Editor and the three Reviewer for the time spent in revising the paper and the opportunity of improving the clarity and quality of the paper.

We did our best to understand all comments and implement all requests: we answered to all comments and we hope the set of answers shall satisfy all Reviewers.  

Thanks

Reviewer 3

Review Comment

This paper presents a novel sensing design on the Running Prosthetic Feet (RPF) that can collect data of track loads in real time. This design modifes a commercial RPF into a instrumented wearable RBF, namely iRBF, by mounting 3 strain gauge bridges onto the prosthetic foot at different locations. The formulation of force measurement and calibration method are presented.

Comparison with ground platform measured forces during the sprint and the jump show good alignment and accuracy. Especially, the iRBF enabled evaluating the athlete's in-field running technique, designing RPF with appropriate structural properties, and so on, which can be dfficult in traditional measuring setups.

The research is well conducted and this paper is well written. However, there are a few concerns I believe should be addressed or clarifed.

ANSWER: we thank Reviewer #3 for the comment and apprecciation. We tried to address all his/her concerns as much as we could.

Concerns

  1. page 1, line 24-25

The last sentence of abstract, "The potential application of this wearable system in the estimation of determinants of sprint performance are presented", should be rephrased because it contains too many "of" in succession and reads odd.

ANSWER: we thank Reviewer #3 for the comment and suggestion. We reworded the sentence as follows.

The potential applications of this wearable system in estimating  determinants of sprint performance are presented.

  1. on page 3, line 113.

There is no gamma  in "Figure 1.b". Maybe the authors was trying to say "Figure 1.c".

ANSWER: we thank Reviewer #3 for the comment. Indeed, the “gamma” angle was referred to Figure 1.c. We corrected the text accordingly.

  1. on page 10, line 325.

Figure 10 is blurred. Please consider re-plotting the two sub-_gures on

Figure 10.

ANSWER: we thank Reviewer #3 for the comment. We modified Figure 10 as requested. .

  1. on page 13, line 400.

Figure 13 is blurred and linewidth is not consistent with that of Figure 12

and others. Please consider re-plotting the two sub-_gures on Figure 13

(If Table. 2 is all correct).

ANSWER: we thank Reviewer #3 for the comment. We clarified The Figure as requested.

  1. on page 16, line 518-519.

The reported error percentage seems inconsistent with that of Table.2. It

seems the "3,5" in line 518 should be "3", while "12%" in line 519 should

be "13%".

ANSWER: we thank Reviewer #3 for the comment. We corrected the numbers accordingly.

  1. It seems that all the athletes in the reported photos were carrying a back-pack. Are they carrying the weight for the battery and computation de-vices? If yes, this can be a backdrop for now.

ANSWER: we thank Reviewer #3 for the comment. As reported in the Method section, line 272, the added mass was 2,3 kg. At present, we have improved the data logging system with a lighter battery, so that the backpack has become quite lighter.

Round 2

Reviewer 1 Report

The authors improved the manuscript and well responded to my comments. I believe that the revised manuscript is acceptable for publication. Congratulation!